# Representative Prototype with Constrastive Learning for Semi-supenvised Few-shot Classification

## Abstract

Few-shot learning aims to learn novel classes in the dataset with few samples per class, which is a very challenging task. To mitigate this issue, the prior work obtain representative prototypes with semantic embedding based on prototypical networks. While the above methods do not meet the requirement of few-shot learning, which requires abundant labeled samples. Therefore, We propose a new model framework to get representative prototypes with semi-supervised learning. Specifically, we introduces the dataset containing unlabeled samples to assist training the model. More importantly, to fully utilize these unlabeled samples, we adopt conditional variational autoencoder to construct more representative prototypes. Simultaneously, we develop novel contrastive loss to improve the model generalization ability. We evaluate our method on miniImageNet and tieredImageNet benchmarks for both 1-shot and 5-shot settings and achieve better performance over the state-of-the-art semi-supervised few-shot method.

## 1 Introduction

In real life, humans are able to quickly establish awareness of new concepts from just one or a few examples. However, conventional machine learning usually learn with abundant labeled samples to ensure its generalization ability. Actually, obtaining a plentiful of labeled samples is exceedingly hard on account of security and the high cost time and money. Motivated by this, many researchers turn to few-shot learning (FSL). In the field of image classification, FSL means getting better image classification accuracy in a small dataset. Generally, prior knowledge is obtained from the base classes and then applied to the novel classes, which contains a few labeled samples (Fei-Fei et al., 2006) (Wang et al., 2020).

Existing studies on FSL roughly fall into four types. (1) Metric-based method (Koch et al., 2015) (Vinyals et al., 2016) (Zhang et al., 2019b). The type of methods is a space mapping method, which aims to learn a good feature space. In this space, all data is converted into feature vectors, and the feature vectors of similar samples are close, while the feature vectors of dissimilar samples are far, so as to distinguish samples, and the distance usually use Euclidean distance (Snell et al., 2017) or cosine distance Chen et al. (2019a). (2) Optimization-based method. In the meta-learning framework, the method first learns a group of good and potential parameters for the network model with a large number of similar tasks, and then uses this group of parameters as the initial value to train on specific tasks, so as to achieve the convergence effect as long as fine-tuning on the new tasks, such as: (Finn et al., 2017) (Lee et al., 2019) (Fallah et al., 2020). (3) Data augmentation-based method (Alfassy et al., 2019) (Schwartz et al., 2018). The fundamental problem of FSL is that samples is few, so it can be solved by increasing the diversity of samples. For example, (Zhang et al., 2019a) proposed to segment the image into foreground and background, and then combine the foreground and background of different pictures, so as to expand the dataset. (4) Semantics-based method (Chen et al., 2019b) (Xing et al., 2019) (Li et al., 2020) (Zhang et al., 2021) (Xu & Le, 2022). This method is a recent research hotspot, which is mainly inspired by zero-shot learning (ZSL). This series of methods use semantic information as auxiliary information to enhance classification performance. In some cases, visual information is richer, while in some cases, semantic information is richer (Xing et al., 2019), which explains that fusing cross-modal information plays an important role in constructing representative class prototypes.

Generally, most of these methods are not used alone but integrated, and almost all are based on the meta-learning framework. However, class prototypes based on the meta-learning framework are not representative enough due to the number of samples in support set is few. Therefore, we propose a new model to construct representative class prototypes. For FSL, prototype-based is the typical method. Simply put, prototype-based is to construct a class prototype for each class using support set, and then keep test samples (from query set) close to the class prototype to which they belong and away from the other class prototypes. Prototypical networks (ProtoNet) firstly to tackle FSL (Snell et al., 2017), the basic idea of which is that samples in each class will be mapped to a feature space through neural network, and calculating the mean features of all samples of each class in this space as class prototype. And then there are a lot of work around ProtoNet.

The novel extensions of ProtoNet (Ren et al., 2018) exploits unlabeled samples when constructing prototypes, moreover, this paper makes the precise analysis of the distractor in unlabeled samples. Furthermore, a cosine similarity based prototypical network to select neighbor samples to augment support set (Liu et al., 2020) and training the regression model to restore the biased prototype with the Euclidean distance between the biased prototype and the real prototype (Xue & Wang, 2020), *etc*. The setting of our work is similar to (Ren et al., 2018), the different is that we cluster unlabeled samples first, and then judge the labels of unlabeled samples according to cluster centers and class prototypes. Except that we adopt the generation model to make full use of unlabeled samples.

We discover that the key of the prototype-based method is how to use a few samples to construct a representative class prototype, which is the challenging task on account of few samples or noise samples. The above approaches mentioned above rely solely on visual features for few-shot classification. Recently, inspired by ZSL, some work combined semantic embedding with prototype-based to enhance the performance. (Xing et al., 2019) utilize cross-modal information (visual features and semantic embedding) to generate visual and semantic prototypes and fuse the two prototypes according to different weights. (Zhang et al., 2021) takes attribute features as prior knowledge to complete the biased prototype. (Xu & Le, 2022) first selects the representative samples in the base classes via assuming that the features of each class follow the Gaussian multivariate distribution. Then, conditional variational autoencoder (CVAE) is used to generate representative features with these representative samples, and constructs representative class prototypes with the generate features and the support features in the novel classes. This paper opens up a ideas for constructing representative class prototypes, the one is data preprocessing, and the other is using generation models to augment data.

Many previous methods require a large number of labeled samples at training stage, which is not consistent with the common scenario in our life, so semi-supervised learning can be used in our work. In order to make full use of unlabeled samples, these samples and semantic information are inputed CVAE to generate more features to construct representative class prototypes. Concurrently, the novel contrastive loss is introduced, which improves model generalization ability. Please note that this contrastive loss is calculated in feature space via feature extractor. Based on the above contents, we propose a novel model framework via meta-learning. Our main contributions of this paper can be summarized as follows:

- We propose a new prototype recovery framework based on meta-learning, which can effectively use unlabeled samples to construct representative class prototypes.

- We develop novel contrastive loss, using class prototypes as the anchor, which allows better inter-class discriminability to mitigate generalization problem.

- We evaluate our approach on two benchmark datasets for few-shot learning, namely mini-ImageNet and tieredImageNet. The experimental results show that our method achieves higher performance, outperforming semi-supervised few-shot learning baselines.

We summarize related works in Section 2. Section 3 provides a rundown of our approach. Section 4 reports the main results obtained with our method. In section 5, we analyzed our methods from different aspects.

## 2 RELATED WORK

### 2.1 SEMI-SUPERVISED LEARNING

Many methods for FSL are supervised learning. However, in many fields, such as medical treatment and aerospace, there will be abundant information that has not been labeled, some of which are useful, but if all of them are manually labeled, it will be very time-consuming and laborious. Therefore, FSL based on semi-supervised learning paradigm is more practical (Ren et al., 2018) (Liu et al., 2019) (Liu et al., 2020). Semi-supervised few-shot learning mainly improves performance with unlabeled samples. Semi-supervised paradigm is used firstly to tackle FSL in (Ren et al., 2018), which augmenting data by tagging unlabeled samples. Simultaneously, they proposed soft K-means, soft K-means + cluster and soft K-means + mask to get more representative prototypes based on ProtoNet. (Liu et al., 2019) puts forward a new idea, which uses graph model and label propagation to predict the labels of query set. Although these methods have achieved good performance, they do not make further use of unlabeled samples, we propose to take these unlabeled samples as the input of CVAE, combined with the corresponding class attributes, to construct more representative prototypes and improve the classification performance.

### 2.2 CONSTRASTIVE LEARNING

The core idea of contrastive learning is to shorten the distance between the positive samples and the anchor sample in the vector representation space, and lengthen the distance between the negative samples and the anchor sample. This makes the boundary between positive and negative samples more obvious. The performance of contrastive learning has been demonstrated in the image domain (Kipf et al., 2019) (He et al., 2020) (Chen et al., 2020) (Han et al., 2021). Compared with the generative methods, which need to reconstruct pixel details to learn sample features, contrastive methods only need to learn discrimination in the feature space. Therefore, contrastive methods do not pay too much attention to pixel details, but can focus on abstract semantic information (more general knowledge), so as to improve the generalization ability of models. For sample selection, in traditional contrastive learning, we regard the input samples as anchors, the samples after data augmentation as positive samples, and other samples in the same batch as negative samples. Differently, sample selection in our model is to use the original prototype as the anchor.

### 2.3 CONDITIONAL VARIATIONAL AUTOENCODER

VAE (Kingma & Welling, 2013) belongs to the generation model family. Using VAE models for generating features conditioned on the corresponding semantic embedding is fairly common in ZSL methods (Mishra et al., 2018) (Schonfeld et al., 2019). (Mishra et al., 2018) is the first to propose to use a conditional VAE for ZSL. (Xu & Le, 2022) is the first FSL method that uses a conditional VAE model to generate visual features, conditioned on the semantic embedding of each class. In (Xu & Le, 2022), focusing on generating more features for the novel classes and constructing representative class prototypes with the support features in the novel classes, while our method focuses on generating more features for the support set to construct the representative class prototypes at two stages: meta-learning training and meta-learning test.

## 3 SEMI-SUPENVISED FSL WITH REPRESENTATIVE PROTOTYPE

In this section, we introduce the overall framework of our model. As shown in Figure 1. For N-way K-shot image classification, we have two set: one is support set $\mathcal{S} = \{(x_i, y_i)\}_{i=1}^{N \times K}$, where $x_i$ is the image, $y_i$ is the label of $x_i$, has few labeled samples. The other is query set $\mathcal{Q} = \{(x_i)\}_{i=1}^{Q}$, where $Q$ is the number of images in $\mathcal{Q}$, contains unlabeled samples. The samples in both sets are from $\mathcal{D}_{novel}$. Samples in $\mathcal{D}_{novel}$ are few, so most work introduce an auxiliary dataset $\mathcal{D}_{base}$. During Meta-Training stage, we add an additional dataset $\mathcal{R} = \{(x_i)\}_{i=1}^{R}$, contains a large of unlabeled samples. Noted that the classes in $\mathcal{D}_{base}$ and $\mathcal{D}_{novel}$ are different, formalized as $\mathcal{C}_{base} \cap \mathcal{C}_{novel} = \emptyset$. Our goal is to classify query samples correctly using few labeled samples from the support set.

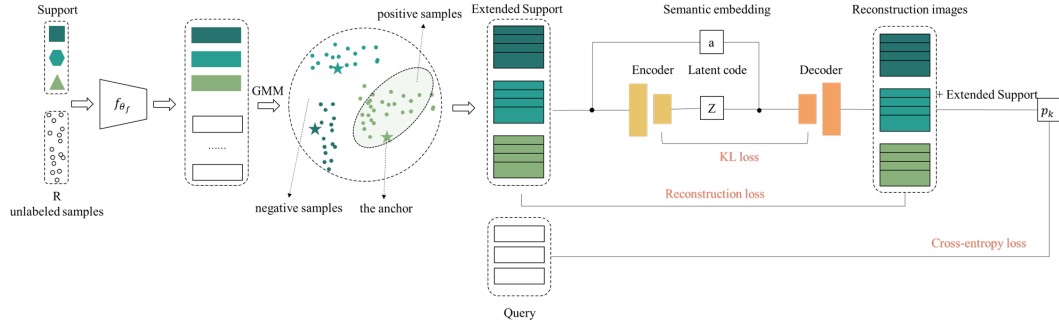

Figure 1: **Overall Framework**. First, getting features representation of samples. Then labeling samples with GMM, the support set turn to the extended support set with unlabeled samples. Finally, constructing class prototypes with CVAE. Training the model with loss function $\mathcal{L} = \mathcal{L}_{cl} + \mathcal{L}_{vae} + \mathcal{L}_{test}$, where the loss $\mathcal{L}_{cl}$ according to shrink the distance of the anchor and the positive samples, while enlarging the distance between the anchor and the negative samples.

In this paper, we exploit a convolutional neural network $f_{\theta_f}$ to extract features from an input $x_i$, where $\theta_f$ indicates a parameter of the network. We adopt ResNet12 as feature extractor for both the support set $\mathcal{S}$ and the query set $\mathcal{Q}$.

## 3.1 LABELING SAMPLES WITH GAUSSIAN MIXTURE MODEL

In this part, generating label for unlabeled samples by the Gaussian mixture model. Specifically,

First, defining a multivariate Gaussian mixture model with K classes to estimate the probability density of the samples. Its defined distribution as Equation (1), where $\mu_k$, $\Sigma_k$ denotes the mean and covariance of the $k$-th multivariate Gaussian distribution, and $\alpha_k$ denotes the probability of the $k$-th mixture component.

$$P(x) = \sum_{k=1}^{K} \alpha_k \cdot p(x|\mu_k, \Sigma_k) \tag{1}$$

Next, calculating the parameters $\Theta = \{\alpha_1, \alpha_2, \ldots, \alpha_k, \mu_1, \mu_2, \ldots, \mu_k, \Sigma_1, \Sigma_2, \ldots, \Sigma_k\}$ with the EM algorithm, which is divided into two steps: E-Step: calculating the posterior probability that the sample $x_j$ comes from the $k$-th multivariate Gaussian distribution, where $R$ denotes the number of $\mathcal{R}$, formalized as Equation (2):

$$\gamma_{jk} = \frac{\alpha_k \cdot p(x_j|\mu_k, \Sigma_k)}{\sum_{k=1}^{K} \alpha_k \cdot p(x_j|\mu_k, \Sigma_k)} \ , \quad j = 1, 2, 3, ..., R \tag{2}$$

M-step: calculating a new round of model parameter estimates ($\alpha_k$, $\mu_k$, $\Sigma_k$), formalized as Equation (3) $\sim$ (5). Repeating the E-step and M-step until the model converges.

$$\alpha_k = \frac{\sum_{j=1}^{R} \gamma_{jk}}{R} \tag{3}$$

$$\mu_k = \frac{\sum_{j=1}^{R} \gamma_{jk} \cdot x_j}{\sum_{j=1}^{R} \gamma_{jk}} \tag{4}$$

$$\Sigma_k = \frac{\sum_{j=1}^{R} \gamma_{jk}(x_j - \mu_k)(x_j - \mu_k)^T}{\sum_{j=1}^{R} \gamma_{jk}} \tag{5}$$

Finally, utilizing the labeled samples from $\mathcal{S}$ to label unlabeled samples. Specifically, after the model convergence is completed, $\mathcal{R}$ is clustered into $K$ classes, expressed as $C = \{c_1, c_2, \ldots, c_k\}$. The model calculates the mean vector as the class prototypes $p_k$ in the support set according to Equation (6), so we have $K$ class prototypes, expressed as $P = \{p_1, p_2, \ldots, p_k\}$. Then the probability of each

cluster $c_i$ to be class $k$ is estimated based on the proximity between the $c_i \in C$ and $p_k \in P$ over a softmax (Bridle, 1990), formalized as Equation (7).

$$p_k = \frac{1}{|\mathcal{S}_k|} \sum_{(x_i, y_i) \in \mathcal{S}_k} f_{\theta_f}(x_i) \tag{6}$$

where $\mathcal{S}_k \in \mathcal{S}$ is the subset of support belonging to class $k$.

$$P(c_i = k) = \frac{exp(-d < c_i, p_k >)}{\sum_k exp(-d < c_i, p_k >)} \tag{7}$$

where $d <, >$ denotes the Euclidean distance of two vectors.

## 3.2 Contrastive Loss

Contrastive learning has been widely shown to capable of improving model generalization. Hence, we develop a new loss to further improve model generalization. Given the class prototype $p_k$ as the anchor, simultaneously, we've possessed the dataset $T = \{x_1, x_2, \ldots, x_t\}$, having the sample label as $p_k$ (in section 3.1), which as the positive samples. Also, having a negative dataset $O = \{x_1, x_2, \ldots, x_o\}$, which has different label from $p_k$. Here we propose a prototype-based contrastive loss function, which adopts the class prototype the anchor, formalized as Equation 8. Now, our goal is to shrink the distance between $x_i \in T$ and $p_k$ while enlarging the distance between $x_i \in O$ and $p_k$.

$$\mathcal{L}_{cl} = -log \frac{\sum_{i=1}^{t} exp(-d < x_i, p_k >)}{\sum_{i=1}^{t+o} exp(-d < x_i, p_k >)} \tag{8}$$

where $t$ denotes the number of positive samples, $o$ denotes the number of negative samples, and $d <, >$ denotes the Euclidean distance of two vectors.

## 3.3 Constructing Class Prototypes with CVAE

Based on the above work, we already have the labeled samples (defined dataset $\tilde{\mathcal{S}}$) according to the support set $\mathcal{S}$ and the corresponding dataset $C$. To generate samples with semantic embedding, we combine $x_i \in \tilde{\mathcal{S}}$ and the interrelated attributes embedding as the input of VAE. VAE consists of an encoder $E(x, a)$, which encodes a sample $x$ to a latent code $z$, and a decoder $D(x, a)$, which reconstructs $x$ from $z$. The objection loss can be defined as:

$$\mathcal{L}_{vae} = KL(q(z|x_i, a^k)||p(z|a^k)) - \log p(x_i|z, a^k) \tag{9}$$

where $a^k$ denotes the semantic embedding of class $k$. The first term is the regularization term, which aligns the variation term $q(z|x, a)$ to the prior distribution $p(z|a)$ through the Kullback-Leibler divergence. The second term is the reconstruction loss, which aims to make the features $z$ from the encoder generated by the decoder approximate the original input features..

We acquire a new dataset $\hat{\mathcal{S}}$ whose samples contain semantic information. Combining the previous datasets $\tilde{\mathcal{S}}$, we can reconstruct representative class prototypes as follows:

$$p_k = \frac{1}{|\tilde{\mathcal{S}}_k|} \sum_{(x_i, y_i) \in \tilde{\mathcal{S}}_k} (f_{\theta_f}(x_i)) + \frac{1}{|\hat{\mathcal{S}}_k|} \sum_{(x_i, y_i) \in \hat{\mathcal{S}}_k} (f_{\theta_f}(x_i)) \tag{10}$$

where $\tilde{\mathcal{S}}_k \in \tilde{\mathcal{S}}$ is the subset of support belonging to class $k$. $\hat{\mathcal{S}}_k \in \hat{\mathcal{S}}$ is the subset of support belonging to class $k$.

## 3.4 Meta-Learning with Representative Ptototype

We construct a number of N-way K-shot tasks from $\mathcal{D}_{base}$ following the episodic training manner (Vinyals et al., 2016). Specially, in each task, we sample $N$ classes from the base classes, $K$ images in each class as the support set $\mathcal{S}$, and $Q$ images as the query set $\mathcal{Q}$. Beyond that, we introduce $\mathcal{R}$, having $R$ images. For each task, first, we access more labeled samples from $\mathcal{R}$ with GMM, and then calculating contrastive loss with these labeled samples, finally constructing class prototypes via

VAE with semantic embedding. Next, calculating the Euclidean distance between the test samples, which from the query set, and the representative class prototypes to predict the labels:

$$P(\tilde{y}_i = k|x_i) = \frac{exp(-d < f_{\theta_f}(x_i), p_k >)}{\sum_k exp(-d < f_{\theta_f}(x_i), p_k >)} \tag{11}$$

Here, $\tilde{y}_i$ denotes the final predicted label for $x_i$, and $d <, >$ denotes the Euclidean similarity between of two vector. The largest probability value is taken as the predict label. Then the loss function is calculated as:

$$\mathcal{L}_{test} = ce(\tilde{y}, y) \tag{12}$$

Here, $\tilde{y}_i$ denotes the final predicted label for $x_i$, $y$ denotes the real label for $x_i$, $ce(,)$ denotes the cross-entropy loss. Therefore, at the meta-learning training stage, the loss function as follow:

$$\mathcal{L} = \mathcal{L}_{cl} + \mathcal{L}_{vae} + \mathcal{L}_{test} \tag{13}$$

Finally, applying the meta-trained model to the novel class.

Table 1: Few-shot classification accuracies on miniImageNet and teiredImageNet. All results are averaged over 600 test episodes. Top results are highlighted.

| Method | Backbone | miniImageNet | | teiredImageNet | |
|---|---|---|---|---|---|
| | | 5-way 1-shot | 5-way 5-shot | 5-way 1-shot | 5-way 5-shot |
| ProtoNet (Snell et al., 2017) | ConvNet-64 | $49.42 \pm 0.78$ | $68.20 \pm 0.66$ | - | - |
| MAML (Finn et al., 2017) | ResNet-18 | $48.70 \pm 1.84$ | $63.11 \pm 0.92$ | - | - |
| Soft k-Means (Ren et al., 2018) | ConvNet-64 | $50.09 \pm 0.45$ | $64.59 \pm 0.28$ | $51.52 \pm 0.36$ | $70.25 \pm 0.31$ |
| TPN-semi(Liu et al., 2019) | ConvNet-64 | 52.78 | 66.42 | 55.74 | 71.01 |
| PSN, Semi-supervised (Simon et al., 2018) | ConvNet-64 | - | $68.12 \pm 0.67$ | - | $71.15 \pm 0.67$ |
| Ours (Semi-supervised learning) | ResNet-12 | $53.10 \pm 0.03$ | $68.14 \pm 0.02$ | $57.14 \pm 0.14$ | $74.13 \pm 0.02$ |

## 4 EXPERIMENTS

### 4.1 BENCHMARKS

**miniImageNet.** In 2016, Google DeepMind team extracted miniImageNet based on ImageNet, which contains 100 classes, each class has 600 images. DeepMind team first applied miniImagenet to few-shot learning research (Vinyals et al., 2016), since then miniImageNet has became the benchmark for few-shot learning fields. In general, we divided the dataset into 64 classes as training sets, 16 as validation sets, and 20 as test sets.

**teiredImageNet.** The data set is also a subset of ImageNet, which proposed in (Ren et al., 2018). teiredImageNet is similar to Omniglot, its classification has the concept of hierarchy. The dataset is divided into 34 high-level classes (such as Instruments, tools, vehicles, *etc.*), each of which contains 10-30 more detailed sub-classes (such as Musical Instruments, including guitars, pianos, *etc.*). The 34 classes are divided into 20 training classes, 6 validation classes and 8 test classes.

### 4.2 IMPLEMENTATION DETAILS

**Dataset Split.** For each dataset, we first create an additional split to separate labeled and unlabeled samples. For miniImagenet, 40% of the data for labeled samples and 60% of data for unlabeled samples. For teiredImagenet, 10% of the data for labeled samples and the remaining 90% for unlabeled samples.

**Training Details.** We adopt ResNet12 architectures as feature extractor, and the dimension of the feature representation is 512. The dimensions of semantic embedding are set to be 512, and are extracted from CLIP Radford et al. (2021). And then, all parameters are trained jointly with 100 episodes in a epoch. All our models are trained with Adam KingaD (2015) and initial learning rate of $10^{-3}$.

Figure 2: **Clustering Visualization. (a)** To update the parameters of ResNet12. **(b)** To compare features distribution of the original and the clustered.

**Evaluation.** We conduct few-shot classification on 600 randomly sampled episodes from the test set and report the mean accuracy together with the 95% confidence interval. In each episode, we randomly sample 15 unlabeled images per class for augmenting the support set, and sample 15 query images per class for evaluation in 5-way 1-shot/5-shot tasks .

## 4.3 RESULTS

Table 1 presents the 5-way 1-shot and 5-way 5-shot classification results of our methods on miniImageNet and tieredImageNet in comparision with previous FSL methods. We compare our method with (Snell et al., 2017) (Finn et al., 2017) (Ren et al., 2018) (Liu et al., 2019) (Simon et al., 2018). The first is classical meta-learning methods, ProtoNet and MAML. The second is Semi-supervised method, such as Soft k-Means and TPN-semi. Since we use the Semi-supervised method paradigm for training, we choose these two methods for comparison. Our method outperforms existing semi-supervised methods, which demonstrates its effectiveness.

## 5 ANALYSES

In order to better illustrate the experimental results, we explained and visualized some details of the experiment, found the experimental results under different experimental settings, and found better experimental settings.

### 5.1 FEATURE DISTRIBUTION ANALYSIS

**Clustering Visualization.** As shown in Figure 2(a), we first extract features of the input samples to obtain 512-dimensional feature vectors. And then in feature space, clustering features of unlabeled samples. Finally, updating the parameters of feature extractor according to the classifier. To sum up, we cluster unlabeled samples in feature space. To further understand the distribution of clusters in the feature space. We use t-SNE (Hinton & van der Maaten, 2008) to map feature representation to 2-d space. Figure 2(b) shows the feature distribution. The left is the original feature distribution, and the right is the clustered distribution. We find that clustering will change the feature distribution of the original data and cluster the features of the same samples. However, since the image is high-dimensional data, the label generated by clustering may not be correct, which also causes problems in downstream tasks. Therefore, we will study how to further improve the accuracy of clustering by GMM.

**Features Visualization on Training.** To verify whether feature extractor has learned discriminative features, we randomly select three classes to visualized the feature distribution at different training stages. Figure 3(a) is the original features distribution. Figure3(b) is the features distribution after epoch 1. Figure.3(c) is the features distribution after epoch 64. Figure3(d) is the features distribution after epoch 128. As shown in Figure 3, we find that with the increase of training epochs, the features extracted by feature extractor become more and more discriminative. This shows that the model obtained in this training stage is effective.

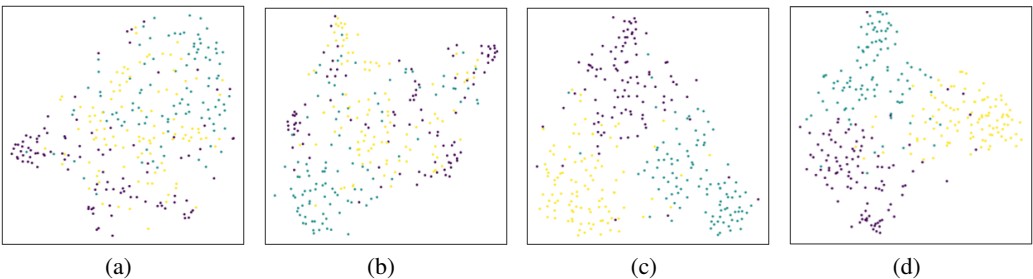

|  (a)  |  (b)  |  (c)  |  (d)  |

Figure 3: **Feature Visualization on Training. (a)** Epoch 0. **(b)** Epoch 1. **(c)** Epoch 64. **(d)** Epoch 128.

## 5.2 Performance on Different number of unlabeled samples

Figure 1 shows the classification accuracy with different number of unlabeled samples. The model is carried out under the setting of $way = 5$. The number of unlabeled samples of each class increases from 0 to 25, and we observe that the classification accuracy of the model also increases. This demonstrates that under the meta-learning framework, even under the semi-supervised paradigm, the model can learn to obtain a better representation.

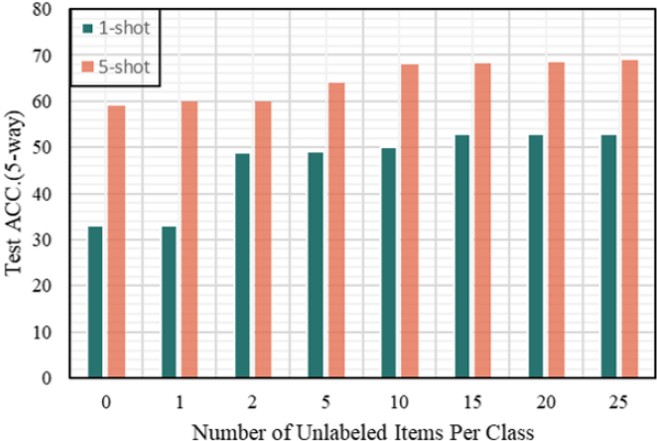

Figure 4: Model Performance on miniImageNet with different numbers of unlabeled samples at meta-learning test.

## 6 Conclusions

A novel framework for few-shot learning to construct representative class prototypes is proposed. First, clustering unlabeled samples with GMM, and predict the labels of unlabeled samples according to the support set with few labeled samples, thus the model conforms to the supervised learning paradigm. Then, in order to learn more distinctive features, the prototype-based contrastive loss is developed. Finally, the corresponding semantic embedding of the extended support set and the extended support set as the input of CVAE to reconstruct more features, so as to calculate more representative class prototypes. The whole model is based on the meta-learning framework, and the whole training loss is composed of three parts. The results show that the performance of our method is better than the existing semi-supervised learning methods. In the future, we will focus on how to get better clustering results in high-dimensional data.

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
