# OpenReview forum: "REPRESENTATIVE PROTOTYPE WITH CONSTRASTIVE LEARNING FOR SEMI-SUPENVISED FEW-SHOT CLASSIFICATION"
_ICLR.cc/2023/Conference — Submitted to ICLR 2023_

### Official Review · Reviewer_7ZBH · 2022-10-21

**Confidence:** 5
**Correctness:** 2
**Technical Novelty And Significance:** 1
**Empirical Novelty And Significance:** 1
**Recommendation:** 1

**Clarity, Quality, Novelty And Reproducibility:**

Clarity, Quality
---
The writing of the paper should be significantly enhanced to reach an acceptance level. There are many grammar issues (singular/plural, lack of phrase structure) and typos (e.g. 'objection loss': 'objective loss', 'teiredImageNet': 'tieredImageNet', ....) that it interferes with the reading and understanding of the paper.

The introduction could be lighten by transfering some parts in the related work section.

Several sections should be improved to better understand what is done or to better justify the use of some techniques. For example:
1. What are the benefits of using a CVAE here ? The loss defined by equation 9 is the classical loss of CVAE; explanations could be omitted to rather focus on the interest of using CVAE here
2. in the explanation of equation 10, the definitions of $\tilde{\mathcal{S}}_k$ and $\hat{\mathcal{S}}_k$ are exactly the same! This contributes to the confusion we have when reading this  paper.

Novelty
---
As discussed in the weakness section, there are no reference to previous work using transductive settings. Although the experimental settings may be considered as pretty close, and that performance reported in transductive settings are far better than the performance reported here.

Reproductibility
----
There are minimal information about the configuration settings used here. And further from the results reported, questions araised about the underlying implementation:
1. confidence intervals reported in Table 1: How can we have 0.03 confidence intervals with only 600 runs where in literature we rather see confidence intervals $\sim 0.2$ while using 10 000 runs!
2. when looking at figure 4 results, the configuration case where we use 0 additional unlabeled samples indeed corresponds to classical inductive settings. For instance, we could the compare the performance here with the one of e.g. MAML... However reported performance are drastically below which let me think that the underlying implementation is having significant problem; not even replicating previous existing results!


**Strength And Weaknesses:**

Strength:
---
this paper illustrates that using semi-supervised learning, some gain may be obtained relatively to inductive settings

Weakness:
---
1. This paper is clearly lacking of recent works dealing with transductive setting in the context of FSL. Indeed transductive settings directly exploit the unlabelled samples from the query set to help improve the design of the classifier without having to introduce the $\mathcal{R}$ dataset which is questionable. So the proposed 'semi-supervised' setting should be clearly compared to this transductive setting.
2. The benefirs of the use of CVAE is not clear here in the paper
3. Experiments are really weak and more datasets should be considered here.
4. Performance reported are really poor with respect to typical performance reported in transductive settings (see [1-3] for early work example on transductive settings). Especially [3] is already exploiting soft clustering of unlabelled samples to improve FSL
5. Feature distribution analysis in section 5.1 is very naive. Comparison with epoch 0 of the training is somehow useless, since the proposed features are rather random ones. A typical instructing comparison would be to compare with final converged solution when using various losses or use of CVAE or not.
6. equation 10 is incorrect when defining prototypes. The propotypes from the support and extended unlabelled sets should be (weighted-)averaged rather than sumed.

Some early references to be considered here about transductive settings: (although they are not the most recent ones)

[1]. Ziko et al, "Laplacian regularized few shot learning", ICML2020

[2] Lichtenstein et al, "TAFSSL: Task adaptive feature sub-space learning for few shot classification', ECCV 2020

[3] Kye et al, "Meta learned confidence for few shot learning", CVPR 2020



**Summary Of The Paper:**

This paper deals with the problem of Few Shot learning were too few samples are available to perform heavy training of the model. To alleviate this problem authors proposed to use semi-supervised learning. An additional set of unlabelled samples is thus introduced. This additional set of samples is exploited with GMM clustering performed on the set of features and further associated to support classes with a specific metric in the latent space of a Contextual Variational Auto Encoder. Those pseudo-classified additional samples are then used to update class protoypes.
All the network used (feature extractor, CVAE) are trained with episodic training and dedicated losses (constrastive loss for prototype, CVAE loss, cross entropy for accuracy efficiency).
Experiments are carried on miniImageNet and tieredImageNet data sets and compared with classical approaches. Additional analyses are provided to illustrate obtained clusters and to show the influence of introducing the additional unlabelles samples.

**Summary Of The Review:**

This paper propose to use semi-supervised learning to enhance Few Shot learning scenarios. However the proposed approach turns out to be very similar to the transductive setting used in Few Shot Learning. No reference to such transductive setting is made here which is mandatory here. Furthermore reported results are far away from those obtained by even very early works in transductive settings.

Further considering the low clarity and quality of the paper, this paper is not of sufficient quality to be accepted here. So I recommend to strongly reject this paper.

---

### Official Review · Reviewer_R3n9 · 2022-10-27

**Confidence:** 5
**Correctness:** 1
**Technical Novelty And Significance:** 1
**Empirical Novelty And Significance:** Not applicable
**Recommendation:** 1

**Clarity, Quality, Novelty And Reproducibility:**

Clarity: Poor

Quality: Poor

Novelty: Poor

Reproducibility: Ok.

**Strength And Weaknesses:**

Strength:
- The method is simple.

Weaknesses:
- The paper was written in a rush with many typos, unfinished sentences, and wrong mathematical formulas.
- The motivation is weak. It's unclear what is the main idea. Using unlabeled data is common in transductive settings. The contrastive loss here is not new.
- The novelty is weak. Using GMM to label unlabeled data is a common practice. Using CVAE to generate additional features is not new. Constructing representative prototypes is not new.

**Summary Of The Paper:**

The authors propose a method for few-shot classification via constructing representative prototypes. Here additional unlabeled data is used (transductive setting). They first use GMM to label samples in an EM manner. A conservative loss is used to push class centers far away from each other. A conditional VAE is trained to generate new samples that are used together with few-shot samples to construct class prototypes.

**Summary Of The Review:**

The paper doesn't introduce novel ideas and is poorly presented.

---

### Official Review · Reviewer_xsww · 2022-11-03

**Confidence:** 4
**Correctness:** 2
**Technical Novelty And Significance:** 1
**Empirical Novelty And Significance:** 1
**Recommendation:** 3

**Clarity, Quality, Novelty And Reproducibility:**

- Clarity: the paper needs significant editing -- including a thorough grammar check, spell-check (even the TITLE has a typo "Supenvised").
- Quality: The results section is lacking in insight -- experiments and analyses should be much more exhaustive than presented.
- Novelty: Limited.  Many of the components (CVAE, prototype construction and using unlabeled data for training) have been used before for FSL
- Reproducibility: no code available

**Strength And Weaknesses:**

*Strengths*
- The topic of FSL is important finding effective prototypes is important for improving FSL.
-

*Weaknesses*
- Many details about the experimental setting are missing. For example how were the dataset splits obtained? Were they sampled at random, or was another protocol used?
- tieredImageNet is an interesting dataset to use, but the paper does not explore various few-shot settings that are possible in tieredImageNet. For example, splitting the dataset on high-level vs low-level.
- No ablation studies on the different components on the method (Contrastive Loss, CVAE protypes, Meta Learning, etc.)
- only one architecture is used. Experiments should be replicated on various architectures (eg. ViT, ResNet, etc.)

**Summary Of The Paper:**

This paper proposes a new method for few-shot image classification. The method contains a new framework for obtaining representative prototypes, by leveraging an external unlabeled dataset to assist training. The method also includes a conditional variational autoencoder and a contrastive loss. Experimental results are shown for 1-shot and 5-shot settings on two datasets (miniImageNet and tieredImageNet).

**Summary Of The Review:**

The paper addresses an important problem but falls short in terms of experimental and analytical insights.  There are several modules in the method, but no ablation study is provided to establish the utility of each module.

---

### Decision · Program_Chairs · 2023-01-20

**Decision:**

Reject

**Justification For Why Not Higher Score:**

A clear reject.

**Justification For Why Not Lower Score:**

N/A

**Metareview: Summary, Strengths And Weaknesses:**

A clear reject. No responses were given by the authors to address significant concerns raised by the reviewers.

**Summary Of Ac-Reviewer Meeting:**

N/A